# Caregiver burden in Buruli ulcer disease: Evidence from Ghana

**Yaw Ampem Amoako**[1,2,3]*, **Nancy Ackam**[1], **John-Paul Omuojine**[3], **Michael Ntiamoah Oppong**[1], **Abena Gyawu Owusu-Ansah**[1], **Mohammed Kabiru Abass**[4], **George Amofa**[5], **Elizabeth Ofori**[6], **Michael Frimpong**[1,2], **Freddie Bailey**[7], **David Hurst Molyneux**[7], **Richard Odame Phillips**[1,2,3]

**1** Kumasi Centre for Collaborative Research in Tropical Medicine, Kwame Nkrumah University of Science and Technology, Kumasi, Ghana, **2** School of Medicine and Dentistry, Kwame Nkrumah University of Science and Technology, Kumasi, Ghana, **3** Komfo Anokye Teaching Hospital, Kumasi, Ghana, **4** Agogo Presbyterian Hospital, Agogo, Ghana, **5** Dunkwa Government Hospital, Dunkwa, Ghana, **6** Tepa Government Hospital, Tepa, Ghana, **7** Department of Tropical Disease Biology, Pembroke Place, Liverpool School of Tropical Medicine, Liverpool, United Kingdom

* yamoako2002@yahoo.co.uk

## Abstract

### Background

Buruli ulcer disease (BUD) results in disabilities and deformities in the absence of early medical intervention. The extensive role of caregiving in BUD is widely acknowledged, however, associated caregiver burden is poorly understood. In this paper we assessed the burden which caregivers experience when supporting patients with BUD in Ghana.

### Method/ principal findings

This qualitative study was conducted in 3 districts in Ghana between August and October 2019. 13 semi-structured interviews were conducted on caregivers of BUD patients in the local language of Twi. Data was translated into English, coded into broad themes, and direct content analysis approach was used to analyse results. The results show the caregivers face financial, psychological and health issues as a consequence of their caregiving role.

### Conclusion/ significance

This study found significant caregiver burden on family members. It also highlighted the psychological burden caregivers experience and the limited knowledge of the disease within endemic communities. Further research is needed to quantify the caregiver burden of BUD at different economic levels in order to better understand the impact of possible caregiver interventions on patient outcomes.

**Data Availability Statement:** All relevant data are within the manuscript and its Supporting information files.

**Funding:** This publication was produced by the BuruliNox study which is part of the EDCTP2 programme supported by the European Union that is held by ROP (Grant number: 101897 BuruliNox TMA 2016 SF-1509). DHM received financial support from Sanofi-Aventis Groupe (Grant number: AG-18-0001148). The views and opinions of authors expressed herein do not necessarily state or reflect those of EDCTP (http://www.edctp.org) or Sanofi Aventis (https://www.sanofi.com). The funders had no role in study design, data collection and analysis, decision to publish or preparation of manuscript.

**Competing interests:** The authors have declared that no competing interests exist.

## Author summary

Buruli ulcer disease (BUD) is a stigmatizing skin condition caused by the bacteria, *Mycobacterium ulcerans*. The disease results in permanent functional limitations in the absence of early medical intervention. The disabling BUD conditions, financial constraints and frequent hospital visits support the role of caregiving for affected individuals. Caregiver burden is poorly understood although the role and need for caregiving is widely recognised in BUD. This study identified a previously unrecognized burden on the caregivers of BUD patients in 3 endemic districts in Ghana. Specifically, we identified significant financial and psychological pressure on affected families in meeting healthcare related costs and physical care while also providing for their own and other family members' needs. We also highlight the emotional burden experienced by caregivers, their reduced work productivity, the barriers caregivers face in accessing healthcare with BUD patients and the limited support available for caregivers. Our study highlights the serious social consequences of BUD in Ghana. Further quantitative research within different economic regions affected by BUD is warranted to better understand the caregiver burden of BUD.

## Introduction

Buruli ulcer disease (BUD) is a necrotizing skin disease caused by *Mycobacterium ulcerans* and is recognised by the World Health Organization (WHO) as a Neglected Tropical Disease (NTD) [1]. The disease occurs in more than 30 countries worldwide with the majority of cases in rural parts of West African countries including Ghana [2]. Sporadic cases are also reported in many locations within the Americas, Japan and the West Pacific regions [3]. Buruli ulcer presents initially as a painless subcutaneous nodule, plaque or edema which may progress with time into ulcers with necrotic bases and undermined edges [2]. This results in large disfiguring skin ulcers mainly in children aged 5 to 15 years though all age groups can be affected [4]. The establishment of the WHO Buruli Ulcer Initiative in 1998 together with improved diagnosis and management have led to major gains in understanding disease mechanism, however, its mode of transmission remains an enigma [5].

Prior to the use of antibiotics, surgery was the most effective means of treatment. Currently, the standard of treatment is the administration of oral antibiotics (specifically rifampicin and clarithromycin) daily for 8 weeks [6–8]. Additionally, adjunctive therapies like wound care, debridement and physiotherapy can reduce complications especially when lesions develop close to a joint [9]. Early complete antimicrobial administration gives excellent treatment outcomes, with return to quality of life [2]. On the other hand, when medical intervention is delayed or incomplete, permanent functional disabilities and their associated stigma [10] and psychological burden ensues [11–13]. Exacerbating these issues, an important loss of economic productivity often ensues [14–16]; in one Ghanaian study, this amounted to 265 working days lost due to BUD [17].

Nevertheless, BUD is prevalent in rural communities where there is poor access to health care facilities [18]. Affected individuals have to travel long distances to access healthcare. Although cost of BUD treatment in most endemic African countries is free, supplementary expenses such as costs of feeding, transportation, and prolonged hospitalization are a huge economic burden [19–22]. This creates a barrier for patients to seek early medical intervention, thus leading to worsening disfiguration and disability.

The debilitating nature of BUD, the young median age of disease occurrence, the significant financial burden and frequent hospital visits leads to a wide-ranging role for caregivers such as

relatives or loved ones. Per the WHO report on ageing and health, '*a caregiver provides care and support to someone else; such support may include*: *helping with self-care, household tasks, mobility, social participation and meaningful activities; offering information, advice and emotional support, as well as engaging in advocacy, providing support for decision making and peer support, and helping with advance care planning; offering respite services; and engaging in activities to foster intrinsic capacity*'[23]. Whereas caregiving can strengthen family ties, the role also demands committing significant amount of time and energy for long periods and performing tasks that may be physically, emotionally, financially, spiritually and socially daunting. The extent to which caregivers perceive these tasks as challenging can be termed 'caregiver burden'[24]; these two definitions are maintained throughout the study. In a previous study, household cost of caregivers of BUD patients was 8.6 times higher than in those that socially isolated themselves from their BUD affected relatives [25].

A number of studies have documented the impact of caregiver burden on caregivers of patients with chronic skin NTDs namely podoconiosis [24], Lymphatic filariasis [26] and cutaneous leishmaniasis [27,28]. In addition, there has been one qualitative study of BUD caregivers to date, conducted in Benin [29]. In this paper, we for the first time, explored the caregiver burden in persons who provide such support to BUD affected patients in Ghana.

## Methods

### Ethics statement

All participants provided written informed consent. The study was approved by the Committee on Human Research, Publication and Ethics (CHPRE) at the Kwame Nkrumah University of Science and Technology (KNUST) in Ghana (approval number: CHRPE/AP/335/19). Staff involved in the study received training on how to appropriately administer questionnaire and conduct interviews. All the study processes were performed in accordance with the principles guiding research in human subjects as set out in the Declaration of Helsinki [30]. The findings from this study have been reported in accordance with the consolidated criteria for reporting Qualitative research (COREQ) checklist (S1 COREQ Checklist).

### Study procedures

This cross-sectional study was part of a larger qualitative and quantitative study of psychological impact in BUD patients and their caregivers. Socio-demographic data of the study participants such as age, gender, occupation and level of education were collected. Qualitative research methods were used to enable participants to describe their experiences in their own words and to ensure that any unanticipated or culturally relevant answers were not missed. The study was carried out among participants attending BUD clinics at Agogo Presbyterian Hospital in the Asante Akim North district, Tepa Government Hospital in the Ahafo Ano North district (both districts located in Ashanti Region) and Dunkwa Government hospital in the Upper Denkyira East district (Central Region). These districts are endemic for BUD in Ghana and have treatment centres for the management of the disease.

### Participant recruitment and sampling

Participants were enrolled if they were actively in a caregiving role as defined by WHO [23] for a patient with active BUD or a one with a recently healed BUD lesion and were willing to provide consent. Caregivers of former BUD patients (past infection) were defined as those who had provided care for patients who were confirmed with BUD within the past 12 months and had completed antibiotic therapy with healed lesions. A caregiver for active BUD patient was one

who provided care for a patient with confirmed BUD who was still receiving antibiotic therapy and whose lesion had not healed. Persons excluded were those with inability to respond to questions, those having mental health conditions that could confound study results and those not residing with or actively giving care to a patient with BUD. Caregivers were identified by BUD patients who attended the BUD clinic employing purposive sampling technique. Caregivers for BUD patients who met the inclusion criteria during the study period were recruited till saturation was reached. Participants were provided with participant information leaflets informing them of the aims and purpose of the study. This was read and explained in the local language, Twi, to those who were unable to read. Participants were recruited (face-to-face) if they met the inclusion criteria. In all, a total of 13 participants were recruited; five (5) from Agogo Presbyterian hospital, two (2) from Tepa Government hospital and six (6) from Dunkwa Government hospital. No caregiver declined participation or dropped out from the study.

## Data collection

Between August and October 2019, data collection was carried out in the three hospitals when caregivers accompanied their sick relatives for medical care. Data was collected via semi-structured interviews, using topic guides (S1 Text) developed by the study coordinator (NA) and study psychiatrist (JPO). The study team members had meetings to discuss the translated versions of topic guides in detail before recruitment began. The topic guides included open questions around caregiver routine responsibilities, impact of BUD on daily life, available support to caregivers and access to healthcare. Face-to-face interviews were conducted in private consulting rooms at the BUD clinics by the study team and lasted an average of 60 minutes. The interviews were conducted by 2 male medical doctors with several years of clinical and research experience [YAA (MD), JPO (MD)] and 2 female researchers [NA (MPhil), AGO-A (MD)]. The interviewers have been involved in providing care for BUD patients who attend the network of clinics where participants were recruited. The interviews were audio recorded on password-protected devices to ensure they were accurately transcribed, and a high-quality recording was obtained.

## Data analysis

The interviews (conducted in the local language, Twi) were translated and transcribed into English from the audio recordings by the study coordinator (NA). The transcripts were then read, sections of the audio recordings listened to for confirmation of the transcript. About one third of the transcripts were verified by an independent research scientist at KCCR who listened to the audio recording in the Twi language while reading the English transcription in order to establish accuracy of the translations. Participants were also given the opportunity to review the transcripts for accuracy. The interviews were uploaded to NVivo 12 which further aided the coding process. The translated interviews were analysed via direct content analysis [31]. The data was first coded into broad themes by the study psychiatrist (JPO) based on a previous research [24]. The themes identified were support for caregivers, impact on care for other family members, direct burden on caregivers as well as barriers to healthcare. These themes were further sub-themed in order to identify the extent of burden study participants face. The themes identified were revised to fit the data collected [31].

## Results

### Characteristics of study participants

The demographic characteristics of study participants are presented in Table 1. Of the 13 participants, there were 10 females and 3 males; 8 of the participants were above 35 years. All but

**Table 1. Socio-demographic characteristics of caregivers of patients with BUD.**

| Characteristic, n = 13 | Proportion |
| --- | --- |
| Male | 3 |
| Female | 10 |
| **Age** | |
| ≥35 years | 8 |
| <35years | 5 |
| **Occupation** | |
| Farmers | 10 |
| Seamstress/Farmer | 1 |
| Trader | 1 |
| Unemployed | 1 |
| **Marital Status** | |
| Married | 10 |
| Divorced | 1 |
| Widowed | 2 |
| **Household size** | |
| 8–11 | 5 |
| 3–7 | 8 |
| **Educational level** | |
| None | 5 |
| Primary | 4 |
| Secondary | 4 |
| **Relationship of caregiver to patient** | |
| Mother | 7 |
| Father | 3 |
| Wife | 2 |
| Sister | 1 |

one (12 of 13) of the participants were caregivers of former BUD patients whose lesions had healed completely with no functional impairments. One participant was a caregiver to a BUD patient with active category I lesion on the hand. With the exception of one participant who was unemployed, 10 were farmers, 1 was seamstress/farmer and 1 was a trader. Most (10/13) participants were married but 2 were widowed. Most of the caregivers were parents (7 were mothers and 3 were fathers) of the BUD patients. An equal number of participants (4/13) had either primary or secondary education; 5 participants did not have any form of education.

## Areas of impact on caregivers of Buruli ulcer disease patients

The broad thematic areas of impact on caregivers responsible for patients with BUD included barriers to accessing care, burden on caregivers, impact on family members and support for caregivers. Sub themes were further examined.

## Burden on caregivers

**Financial burden.** Caregivers reported experiencing significant financial burden resulting in some cases, in them being required to sell off their land and/or other property. Some were also forced to subsist more on their own farm produce thus reducing their nutritional diversity.

*'During the sickness, I began facing financial difficulties. I had to sell my land in order to raise funds to pay for the upkeep of my ward during that time.'*

*(c3)*

*'Yes, for instance the money spent in bringing her to the hospital continuously. We mostly have to go sell our farm produce in order to get some money to cater for her medically.'*

*(c8)*

*'While he was bedridden, I was the only one working so I could not bring in so much and this put a burden on us. However, as a farmer, I was bringing in some farm produce so we were managing on that.'*

*(c13)*

**Psychological/emotional/ physical health.** The psychological burden of caring for affected patients in this study was expressed by caregivers commonly as 'worry', 'stress' or 'being troubled'.

*I did not have enough money to take care of my children and it brought a lot of psychological stress to me.'*

*(c4)*

*'I became burdened with worrying thoughts during that time.'*

*(c5)*

*'It had also left me with worrying thoughts. . . I also did not know the cause of the disease and this caused me great worry. I would even say that some of those depressing thoughts have led me to develop a constant high blood pressure over time due to the stress.'*

*(c6)*

One caregiver also reported disturbed sleep as a result of her child's condition.

*'. . .how can I sleep when my child is not able to? I get very disturbed when he is doing that.'*

*(c10)*

Another respondent attributed her chronic headaches to the burden of taking care of her husband.

*'Since I came to marry my husband who already had the condition, I cannot say for sure but I have been having chronic headaches which previously were not there.'*

*(c13)*

**Occupational.** Much of the occupational impact of caring for relatives or wards with BUD was expressed in terms relating to lost time and/or output or input at work- mostly farming.

*'. . . As a rice farmer, you always have to be vigilant on the farm against pests such as birds if not they will consume everything you have planted. Since my child was not well, I couldn't say*

*I will turn a blind eye to my child's condition in order to ward off the birds on the rice farm. I had to come with the child every time and by the time I get to the farm, the birds would have destroyed the majority of my crops.'*

*(C10)*

*'I am a farmer and having to accompany my son every time to the clinic to dress his wound has greatly impacted my work output.'*

*(C9)*

*'Yes, I used to work initially but my work output has reduced.'*

*(C14)*

**Social/stigma.** Some respondents reported stigmatizing experiences as a result of the wards' illness.

*'Because of the spiritual causes attached, some people do not even want to associate with us in anyway. Like the way the lesion has shown up on the girl's leg, there are some who will even not want to come close to the child in any way.'*

*(C8)*

**Impact on other family members.** Other family members, usually siblings were affected as their parents had to spend an inordinate amount of time and money attending to their siblings affected by BUD.

*'I was more focused on her during that period when she had BU and could not focus much on her other siblings.'*

*(C6)*

They also suffered the effects of the financial burden that Buruli ulcer brought on the family.

*'Since they were with my mother, she took care of them and fed them. However, I did not have enough money and could not sponsor their education.'*

*(C4)*

*'. . .it affected the education of her two older siblings because I could not finance their schooling.'*

*(c6)*

## Barriers to accessing care

**Financial.** Financial difficulties and perceived cost of care was one barrier to accessing care.

*'Initially, I had the notion that coming to seek health care for my child would be very costly for us so I was finding it difficult to come to the hospital with my ward. Later, one of her older siblings advised me to send her to the clinic. . .'*

*(C6)*

**Transportation.**    Access to transportation appeared to be a significant barrier to accessing care for some respondents.

*'It was difficult getting to the hospital since the roads as at then were quite terrible and there were only a few cars plying the road.'*

*(C1)*

*'there are actually no cars. Only motorbikes serve as our means of transport and they are not readily accessible as you see them once a while. So, you at times have to walk for long before meeting one if you're lucky.'*

*(C9)*

*'Yes that was our main challenge. Sometimes we are fortunate to get a car passing by so we stop it and get on board to the hospital.'*

*(C11)*

## Support for caregivers

**Physical care/ transportation.**    Respondents received support from others (such as siblings) in caring for their wards and transporting them to the hospital.

*'Yes, my elder sister was around and would at times go and take care of her in the afternoon even before I get there. . . in coming to the hospital, we cannot lift her on our own so the men come with us.'*

*(C11)*

**Financial/ material.**    Caregivers reported receiving financial and material support from friends, family, community members and health workers.

*'But his friends came to help by visiting and they gave us money sometimes.'*

*(C5)*

*'They did help out a little. Her older siblings supported me financially when I was taking care of her.'*

*(C6)*

*'With the first one, I had help from my mother since she was alive during that period. She used to give me money for some drugs and food for my ward.'*

*(C4)*

*'Yes, I did get support from my neighbours and friends.'*

*(C6)*

*'We were alone and relied on the money we got from the health workers as transport fare.'*

*(C2)*

*'Also, the first time we came to the hospital after the diagnosis, we were given some money as transport reimbursement and since then we were always reimbursed.'*

*(C10)*

One respondent reported getting a salary advance from their employer.

*'The only help I got was getting salary advance from my boss so I could get enough money to send them.'*

*(C1)*

**Psychological/ emotional.** Friends, family and community members provided emotional/psychological support for caregivers and BUD patients.

*'. . . some of my friends who came to visit my ward at the hospital brought gifts for her.'*

*(C1)*

*'Yes, sometimes his friends do come over to visit him and encourage him to take heart.'*

*(C14)*

**Absence of support.** Some caregivers reported receiving no support whatsoever from others:

*'No, I did not receive such help. Initially, I did not know what was wrong with him so I only sent him to the hospital to get treatment.'*

*(C3)*

*'For my friends, they are all residing up north (which is very far from where we are) so I don't receive any form of help from them.'*

*(C7)*

*'. . . I have not received any form of help. Some foreigners came to the clinic some time ago and examined my son's lesion and asked him series of questions after. However, they did not offer any form of help after'*

*(C9)*

## Discussion

While the psychological impact of BUD on patients has become increasingly recognised [11–13], this is the first qualitative study of caregiver burden in BUD in Ghana, and expands a small but important evidence base for this neglected aspect of BUD. The present study also represents one of only a few studies to look at caregiver burden for any NTD [24,26–29].

Caregiver burden, defined in this study as 'the extent to which caregivers perceive that caregiving has had an adverse effect on their emotional, social, financial, physical, and spiritual functioning [24], was prominent among caregivers of those affected by BUD in three districts in Ghana. We categorized the caregiver burden into four main themes: 1) Personal burden; 2) Impact on other family members; 3) Barriers to accessing care; 4) Support for caregivers.

Within these four themes, we have highlighted the significant financial impact of caring for patients with BUD. The loss of productivity an affected BUD patient faces inevitably places significant pressure on the caregiver to provide for the rest of the family. This is reflected by the finding that almost all of the caregivers in this study (12/13) were employed. Financial burden reported here is in keeping with a previous Ghanaian study where the household costs of caregivers of BUD patients rose by up to 8.6 times [25]. Study participants also reported caregiver burden in terms of reduced occupational output, psychological and health issues. In a study on podoconiosis patients' caregivers in Ethiopia, caregivers provided care for an average of 14.8 and 15.5 days per month for patients with mild/moderate and severe disease respectively. Furthermore, the hardship experienced by family units was not only due to the patients decreased working hours, but, was also due to caregivers' lack of productivity while providing care [32]. Psychological burden reported by caregivers was directly related to the financial pressure they came under as worrying thoughts of not being able to provide for other family members was troubling and corroborated the study on caregiver burden in another disabling skin NTD, podoconiosis [24]. However, in addition to this psychological impact, caregivers also reported the provision of physical care had a negative impact on their own health and well-being. The amount of time dedicated to providing physical care and the extra responsibilities a caregiver needed to assume placed significant stress on the individual, affecting their health. It was also apparent from our study that like BUD patients, caregivers also faced the issue of stigma in their communities. Stigma in BUD is caused by the presence of large disfiguring ulcers coupled with, an unknown mode of disease transmission, leading to associations with cultural and religious beliefs [10,14]. As a result, although patients do still wish to interact within their communities, affected individuals and their families become socially isolated [33,34].

In BUD management, seeking early complete medical intervention lowers the risk of permanent functional limitations and its associated impact on quality of life [2] and psychological co-morbidities [11,12]. Caregivers narrated how finances and non- availability of simple logistic support became barriers to accessing healthcare for their sick relatives. BUD is endemic in rural communities which usually have poor roads and limited geographic access to major health centres. It is therefore unsurprising that caregivers experienced difficulty with transport, leading to delays in seeking early patient care as reported by one caregiver in our study. One solution to the issue of transportation on caregiver burden has recently been highlighted in work surrounding decentralised BUD patient care [35]. This alternative approach has particular benefits to female caregivers, who have been shown to prefer this approach as it facilitates their frequent role in mobilising caregiver resources for BUD patients in Benin [29].

Support available to caregivers greatly influenced the extent a caregiver felt burdened having to cater for sick relatives. The more support a caregiver received, the less likely the person becomes burdened with his role [36]. Some caregivers reported that they received financial support from health service facilities, friends and family. BUD typically affects people of lower socio-economic status and the high cost of management is financially draining on the patient and his family. In Ghana, antibiotics for BUD treatment are distributed for free under the auspices of the National Buruli Ulcer Control Programme with support from the WHO; but patients and caregivers may incur some cost related to care. Hospital care costs may include direct costs like admission fees (when needed as may occur if sepsis develops from secondary wound infection), wound dressings, fees for surgical procedures like debridement and skin grafting (when done), feeding and indirect costs related to loss of economic activity. Most BUD care provided in Ghana is on outpatient basis but patients and caregivers may still have to cater for transportation to and from health facilities, feeding and other ancillary costs. Other caregivers reported that they did not receive any form of support and this greatly affected them.

The Sustainable Development Goals (SDG) target 5.4 [37] seeks to recognise and value un/under paid care and domestic work. Caregivers of patients with chronic NTDs provide unpaid care. The BUD control programme in Ghana should ensure that there are pathways to provide advocacy for support of caregivers of persons with chronic NTDs like BUD. Caregivers should be included in the programmatic planning for management of patients with chronic NTDs like BUD. National BU control programmes should include packages like education and psychosocial support services that will help improve the experience of caregivers of BUD patients.

While this study has made some novel and valuable findings, there were a number of limitations. The study did not include any children who were caregivers. In Ghana, children may provide such support as running errands, helping with farming and cooking among other household chores. Since BUD disease can affect all ages, it would have been interesting to understand caregiver burden from the view point of child caregivers who are offering such support to parents or other older relatives affected by BUD. There was no provision for field notes and documentations of observations made during qualitative interviews. Although we observed and acknowledged some emotions of participants during the interviews, we did not document any of these. Field notes and documentations are subjective measure assessments and not a substitute for the tool employed. No caregivers of chronically disabled BUD patients were included in the study, thus we were unable to assess the impact on caregivers of lasting functional impairment. Furthermore, the assessed caregiver burden could have been impacted by recall bias as most participants were caregivers of former patients. In spite of these limitations, this study has provided valuable information to allow for a more comprehensive approach to managing BUD patients and their caregivers.

## Conclusion

Overall, the debilitating impact of BUD negatively affects not only patients but also their caregivers to a significant degree. Within the context of NTDs, the evidence base for caregiver burden remains restricted to a small number of chronic stigmatising skin NTDs. Therefore, more qualitative and quantitative studies are needed to grow the evidence base in this area. Efforts should be made to provide financial assistance such as waivers on hospital care costs and other incentives for caregivers of BUD patients as this could go a long way to ease the burden of caregiving for BUD in Ghana. Lastly, our study highlights importance of further work on the benefits of integrating psychosocial interventions in NTD management for both patients and their caregivers alike.

## Supporting information

**S1 COREQ Checklist. COREQ checklist for caregiver burden in Buruli ulcer disease: Evidence from Ghana.**
(PDF)

**S1 Text. Topic guide for caregiver burden in Buruli ulcer disease.**
(PDF)

## Acknowledgments

We are grateful to staff of the network of clinics involved in the provision of care for patients with Buruli ulcer disease (BUD) for their assistance. We thank all the caregivers of BUD patients who participated in the study; we appreciate their time and effort in providing care for patients. We thank Jonathan Adjei, a researcher at the Kumasi Centre for Collaborative Research for his assistance with the study.

## Author Contributions

**Conceptualization:** Yaw Ampem Amoako, Nancy Ackam, David Hurst Molyneux, Richard Odame Phillips.

**Data curation:** Nancy Ackam, John-Paul Omuojine, Michael Ntiamoah Oppong, Abena Gyawu Owusu-Ansah.

**Formal analysis:** Yaw Ampem Amoako, Nancy Ackam, John-Paul Omuojine.

**Funding acquisition:** David Hurst Molyneux, Richard Odame Phillips.

**Investigation:** Yaw Ampem Amoako, Nancy Ackam, John-Paul Omuojine, Michael Ntiamoah Oppong, Abena Gyawu Owusu-Ansah, Mohammed Kabiru Abass, George Amofa, Elizabeth Ofori.

**Methodology:** Yaw Ampem Amoako, Nancy Ackam, John-Paul Omuojine, Michael Frimpong, Freddie Bailey, David Hurst Molyneux, Richard Odame Phillips.

**Project administration:** Yaw Ampem Amoako, Michael Frimpong.

**Resources:** David Hurst Molyneux, Richard Odame Phillips.

**Supervision:** Yaw Ampem Amoako, David Hurst Molyneux, Richard Odame Phillips.

**Validation:** Yaw Ampem Amoako, Freddie Bailey, David Hurst Molyneux, Richard Odame Phillips.

**Writing – original draft:** Yaw Ampem Amoako, Nancy Ackam, John-Paul Omuojine.

**Writing – review & editing:** Yaw Ampem Amoako, Michael Ntiamoah Oppong, Abena Gyawu Owusu-Ansah, Mohammed Kabiru Abass, George Amofa, Elizabeth Ofori, Michael Frimpong, Freddie Bailey, David Hurst Molyneux, Richard Odame Phillips.

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
