## [Decision Letter · Decision Letter 0]

24 Mar 2021

Dear Amoako,

Thank you very much for submitting your manuscript "Caregiver burden in Buruli Ulcer Disease: evidence from Ghana" for consideration at PLOS Neglected Tropical Diseases. As with all papers reviewed by the journal, your manuscript was reviewed by members of the editorial board and by several independent reviewers. The reviewers appreciated the attention to an important topic. Based on the reviews, we are likely to accept this manuscript for publication, providing that you modify the manuscript according to the review recommendations. 

Thank-you for your submission and hard work in undertaking this study on Buruli ulcer disease. 

Please note and address the helpful comments detailed by the reviewers below which I concur will improve the clarify of the manuscript.

Sincerely,

Claire Fuller

Guest Editor

Michael Marks

Deputy Editor

Thank-you for your submission and hard work in undertaking this study on Buruli ulcer disease. 

Please note and address the helpful comments detailed by the reviewers below which I concur will improve the clarify of the manuscript.

Reviewer's Responses to Questions

**Key Review Criteria Required for Acceptance?**

**Methods**

-Are the objectives of the study clearly articulated with a clear testable hypothesis stated?

-Is the study design appropriate to address the stated objectives?

-Is the population clearly described and appropriate for the hypothesis being tested?

-Is the sample size sufficient to ensure adequate power to address the hypothesis being tested?

-Were correct statistical analysis used to support conclusions?

-Are there concerns about ethical or regulatory requirements being met?

Reviewer #1: Please see General Comments

Reviewer #2: This study is a qualitive study conducted in Ghana among 13 caregivers of BU patients. Majority of whom had a healed BU lesion. I think the study should be described as such to reflect that the caregivers are caregivers of former BUD. This should be indicated in the title. In addition the study did not include BUD patients with permanent disability. I would expect different perspective from caregivers of patients with permanent disability. Having said that the objectives of the study are clear, the study design is appropriate to respond to the objectives of the study. I have indicated my concern on the appropriateness of the study population for the study. I think this can be adjusted by revising the title of the study to include former BUD cases. The sample size was based on saturation which is appropriate for a qualitive study design. Proper ethical procures were followed.

**Results**

-Does the analysis presented match the analysis plan?

-Are the results clearly and completely presented?

-Are the figures (Tables, Images) of sufficient quality for clarity?

Reviewer #1: Please see General Comments

Reviewer #2: The results are presented appropriately. The analysis was also conducted clearly and following standard methods. My only comment here is since the number of participants is only 13, better to avoid percentage. Presenting the actual number would suffice.

**Conclusions**

-Are the conclusions supported by the data presented?

-Are the limitations of analysis clearly described?

-Do the authors discuss how these data can be helpful to advance our understanding of the topic under study?

-Is public health relevance addressed?

Reviewer #1: Please see General Comments

Reviewer #2: The conclusions of the study are supported by the data presented in the study. The authors have discussed some of the limitations of the study, one example could be the have not discussed that they did not included caregivers of BUD with active lesion, which might have introduced bias to the study. 

The public health relevance of the study is not well addressed. The authors should discuss how the study helps programme implementations.

**Editorial and Data Presentation Modifications?**

Reviewer #1: Please see General Comments

Reviewer #2: (No Response)

**Summary and General Comments**

Reviewer #1: This is an interesting manuscript that expands the relatively small body of literature on the roles, activities and impacts on caregivers of supporting a person affected by a Neglected Tropical Disease (NTD). Amoako and colleagues have conducted a small qualitative study on caregivers of people with Buruli Ulcer disease (BUD) recruited at specific BUD clinics at three hospital sites in Ghana. They analyse data using themes identified in a recent qualitative study of caregivers of people with podoconiosis, another skin-NTD, and describe the financial, psychological, occupational and emotional impacts on the caregivers and other family members.

While overall, this is a useful contribution to the literature, I offer a number of comments that may increase the accuracy of description of the methods, and also improve the Discussion. Several of these arise from discrepancies between what is indicated in the COREQ checklist and what is found in the manuscript.

Methods

Participant recruitment (lines 98-110). Please include a statement as to whether any caregiver declined to participate, or dropped out. This is item 13 of the checklist, which you indicate has been addressed on p7, but I cannot find where.

Line 115. The study psychiatrist is referred to as ‘JP’, but I cannot find a name on the author list whose initials these might be.

Data collection (lines 112-121). You need to add who conducted the interviews, their qualifications and gender (as per items 1-5 on the COREQ checklist).

Line 119. 20-40 minutes does seem short for qualitative interviews based on a topic guide. Please attach the topic guide to your revision to make it easier to assess whether sufficient depth was likely to have been reached.

Data analysis (lines 132-143). There seems to be some confusion between transcription and translation in this para, eg line 132 “Interviews were first transcribed to Microsoft Word in English” – do you mean in Twi? Lines 133-4 “The transcripts were then read…and verbally translated into English..” – this presumably means the transcripts had been in Twi. Presumably the translations were written as well as ‘verbal’? Please clarify this para.

Either under Methods or in the Discussion (as a limitation), you need a statement about the biases and assumptions of the interviewers and those analysing the data (item 8 of the COREQ checklist).

Results

The choice of quotes is good, and there is no undue repetition.

Lines 148-9. Could you clarify this - do you mean caregivers had previously given care to patients whose lesions had healed completely in addition to the patient they accompanied to hospital, or that the patient they accompanied to hospital had a lesion that had healed completely (in which case why did they need to go to hospital)?

Line 182. Figure 1 is very blurred, please improve the resolution. Also add ‘adapted from (24)’.

Discussion

Line 325. Explain why you used these four themes.

Line 333. In discussing reduced caregiver occupational output, you might want to make comparisons with: “Quantifying the socio-economic impact of leg lymphoedema on patient caregivers in a lymphatic filariasis and podoconiosis co-endemic district of Ethiopia” (PLoS NTDs, https://doi.org/10.1371/journal.pntd.0008058)

Line 342. In the results, you say that only one caregiver reported stigmatising experiences, but in the Discussion you say “caregivers also faced the issue of stigma…” - which is correct? If only one caregiver did report this experience, do you think saturation was genuinely reached?

Lines 388-9. While it is good you are suggesting some practical measures, is providing financial assistance for patients really 'simple'?

Minor Issues & Typographic Errors.

Throughout – I prefer ‘data’ to be considered a plural noun (and so take a plural verb), but realise that not all journals take issue with it being used as a singular noun.

Line 92. ‘among’ or ‘with’ would be better than ‘on’.

Lines 114 & 116. ‘topic’ guides, not ‘topical’ guides!

Line 146. ‘The demographic characteristics of study participants are presented..’ (not ‘is’)

Line 187. Should read ‘..resulting in some cases, in them being requiring to sell off...' 

Line 324. Remove second ‘in’.

Line 341. The rest of this sentence uses the past tense, so suggest ‘placed’ rather than ‘places’.

Line 370. Use ‘include’ rather than ‘have’.

Reviewer #2: • The abstract in the online system and in the manuscript are different align this two. 

• The authors indicated that ‘All but one (12 of 13) of the participants were caregivers of former BUD patients whose lesions had healed completely with no functional impairments.’ This should be described properly, how long has been since the lesion had been held. I am asking this because it is obvious that people would forget the burden of the care or at least they will not describe the burden properly as those who are currently caring for BU patient. In addition the duration of when the lesion has been healed would also affect, what the caregivers can remember and report. Therefore, this should be clearly discussed in the context and the implication on the findings should be discussed. 

• Since the number of participants is only 13, I do not think it is proper to use percentage. It would be better to reports the numbers in the text and table 1. 

• Line 366: Can you please describe what ‘the cost of hospital care and transportation’ are? It would be good to estimate the average duration of hospital stay of a BU patient to bring the discussion into context?

PLOS authors have the option to publish the peer review history of their article (what does this mean?). If published, this will include your full peer review and any attached files.

Reviewer #1: No

Reviewer #2: No

Figure Files:

Data Requirements:

Reproducibility:

References

---

## [Decision Letter · Decision Letter 1]

8 May 2021

Dear Amoako,

We are pleased to inform you that your manuscript 'Caregiver burden in Buruli Ulcer Disease: evidence from Ghana' has been provisionally accepted for publication in PLOS Neglected Tropical Diseases.

Best regards,

Claire Fuller

Guest Editor

Michael Marks

Deputy Editor

<style type="text/css">p.p1 {margin: 0.0px 0.0px 0.0px 0.0px; line-height: 16.0px; font: 14.0px Arial; color: #323333; -webkit-text-stroke: #323333}span.s1 {font-kerning: none

</style>

Reviewer's Responses to Questions

**Key Review Criteria Required for Acceptance?**

**Methods**

-Are the objectives of the study clearly articulated with a clear testable hypothesis stated?

-Is the study design appropriate to address the stated objectives?

-Is the population clearly described and appropriate for the hypothesis being tested?

-Is the sample size sufficient to ensure adequate power to address the hypothesis being tested?

-Were correct statistical analysis used to support conclusions?

-Are there concerns about ethical or regulatory requirements being met?

Reviewer #1: All the suggested changes to the methods section have been completed.

Reviewer #2: The authors clearly articulated the objectives of the study, used appropriate methods, with no ethical concern.

**Results**

-Does the analysis presented match the analysis plan?

-Are the results clearly and completely presented?

-Are the figures (Tables, Images) of sufficient quality for clarity?

Reviewer #1: All the suggested changes to the results section have been completed.

Reviewer #2: The results is well resented.

**Conclusions**

-Are the conclusions supported by the data presented?

-Are the limitations of analysis clearly described?

-Do the authors discuss how these data can be helpful to advance our understanding of the topic under study?

-Is public health relevance addressed?

Reviewer #1: All the suggested changes to the conclusions section have been completed.

Reviewer #2: The conclusions are supported by the data.

**Editorial and Data Presentation Modifications?**

Reviewer #1: All minor typos corrected.

Reviewer #2: None.

**Summary and General Comments**

Reviewer #1: The authors have addressed my earlier comments very thoroughly.

Reviewer #2: The authors addressed all my comments and the manuscript is acceptable for publication in PLOS NTDs journal.

PLOS authors have the option to publish the peer review history of their article (what does this mean?). If published, this will include your full peer review and any attached files.

Reviewer #1: **Yes: **Gail Davey

Reviewer #2: No

---

## [Editor Report · Acceptance letter]

20 May 2021

Dear Amoako,

We are delighted to inform you that your manuscript, "Caregiver burden in Buruli Ulcer Disease: evidence from Ghana," has been formally accepted for publication in PLOS Neglected Tropical Diseases.

Best regards,

Shaden Kamhawi

co-Editor-in-Chief

Paul Brindley

co-Editor-in-Chief
